# Unified Perspectives on Signal-to-Noise Diffusion Models

## Abstract

Diffusion models (DM) have become essential components of generative model-ing, demonstrating exceptional performance in domains like image synthesis, au-dio generation, and complex data interpolation. Signal-to-Noise diffusion models represent a broad family encompassing many state-of-the-art models. Although several efforts have been made to explore Signal-to-Noise (S2N) diffusion models from different angles, a comprehensive study that connects these viewpoints and introduces new insights is still needed. In this work, we provide an in-depth per-spective on noise schedulers, analyzing their role through the lens of the signal-to-noise ratio (SNR) and its relationship to information theory. Based on this frame-work, we introduce a generalized backward equation to improve the efficiency of the inference process.

## 1 Introduction

Diffusion models (DM) have become a fundamental part of generative models, which excel in var-ious domains, including creating images, generating audio, and interpolating complex data. The foundational framework for these models was introduced by Sohl-Dickstein et al. (2015), and Ho et al. (2020) further refined it with Denoising Diffusion Probabilistic Models (DDPMs). DDPMs add noise to data iteratively and learn to reverse this process, allowing them to model data distributions effectively.

Signal-to-Noise (S2N) diffusion models (Kingma et al., 2021; Kingma & Gao, 2024) constitute an extensive class of diffusion models encompassing various other models, such as variance-preserving (VP) and variance-exploding (VE) DMs (Song et al., 2020b), iDDPM (Nichol & Dhariwal, 2021), DDPM (Ho et al., 2020), and EDM (Karras et al., 2022). Originally, continuous variation models were introduced by Kingma et al. (2021). They first developed a discrete S2N diffusion model, followed by a variational-based backward inference, and finally examined the asymptotic behavior as the number of time steps approaches infinity, leading to a continuous variational DM. Building on the development of continuous variational DMs, Kingma & Gao (2024) further investigated S2N diffusion models in the signal-to-noise space, identifying connections between diffusion objectives with different weighting formulas and simple data augmentation techniques. An intriguing ques-tion arises regarding whether the forward and backward distributions developed in Kingma et al. (2021) using the variational approach and asymptotic analysis are consistent with and connectable to the forward and backward processes of the Stochastic Differential Equation (SDE) viewpoint of diffusion models (Song et al., 2020b).

Moreover, using the tool developed in Zhang & Chen (2022), we can conveniently derive the back-ward SDE of S2N diffusion models. However, identifying the exact solution of this backward SDE is non-trivial, as it only describes the transition from $z_t$ to $z_{t-\Delta t}$ over a small interval $\Delta t$, rather than allowing us to directly jump from $z_t$ to $z_s$ for $s < t$. It is worth noting that (Zhang & Chen, 2022) addressed the same issue for the ODEs of VPSDE (Ho et al., 2020) and VESDE (Song et al., 2020b). Here, we extend this analysis by providing a solution for the backward SDE of general S2N diffusion models. Additionally, we connect this to the backward transition probability $p_\theta(z_s \mid z_t)$, showing that this is a special case within the spectrum of our developed formulas, which we can exploit to obtain better samples. Additionally, we look into the Non-Markovian continuous vari-ational model of the S2N diffusion models. We note that DDIM (Song et al., 2020a) relaxed the Markov property to arrive in the discrete Non-Markovnian diffusion model for DDPM (Ho et al.,

2020). Here we develop more general Non-Markovnian continuous variational model of the S2N diffusion models. Furthermore, we investigate the backward SDE for Non-Markovnian continuous variational model of the S2N diffusion models which is a novel result because what was gained in Song et al. (2020a) is the connection between DDIM and the probability flow ODE (Song et al., 2020b).

Last but not least, inspired by Kingma et al. (2021); Kingma & Gao (2024), we transform the S2N diffusion models into the signal-to-noise space. Under certain conditions, we find that the original S2N diffusion models, which form an equivalent class, are transformed into the same diffusion model in the signal-to-noise space. Furthermore, we develop an information-theoretic perspective for S2N diffusion models in this space, which can be seen as an extension of the work by Kong et al. (2023), whose analysis was limited to specific and simple S2N diffusion models.

In this work, we propose connective viewpoints of S2N diffusion models. Specifically, our contribution can be summarized as follows:

- We devise a forward SDE for S2N diffusion models and demonstrate its connectivity and consistency with the results developed in (Kingma et al., 2021). Moreover, through asymptotic analysis, we show that we can inversely recover the developed forward SDE from the formula presented in Kingma et al. (2021).

- To enable sampling, drawing inspiration from Zhang & Chen (2022), we devise a general backward SDE and an exact inference formula to transition from time step $t$ to $s$ where $s < t$. Furthermore, we develop a parameterized approximate inference formula for $s = t - \Delta t$. Interestingly, we observe that the inference formula presented in Kingma et al. (2021) aligns with our parameterized approximate inference formula.

- Specifically, drawing inspiration from the Non-Markovian forward process in Song et al. (2020a), we develop a continuous variational diffusion model capable of exactly inducing the forward distributions. Moreover, we devise the backward SDE corresponding to this Non-Markovian inference formula.

- Furthermore, drawing inspiration from Kingma et al. (2021); Kingma & Gao (2024), we map S2N diffusion models onto the signal-to-noise space. Within this framework, we develop an information-theoretic perspective for a general S2N diffusion model, which can be seen as a generalization of the approach presented in Kong et al. (2023).

- Finally, we employ our parameterized approximate inference formula to sample images from existing pre-trained models. This demonstration illustrates that by selecting appropriate parameters, we can achieve higher performance than the inference baselines within the spectrum.

## 2 RELATED WORK

Diffusion models have rapidly become a cornerstone in the landscape of generative models, demonstrating exceptional capabilities across a variety of domains, including image synthesis, audio generation, and complex data interpolation. The foundational framework of diffusion probabilistic models was first introduced by Sohl-Dickstein et al. (2015), and this framework underwent significant refinement with Ho et al. (2020), who developed Denoising Diffusion Probabilistic Models (DDPMs). DDPMs iteratively add noise to data and learn to reverse this process, effectively modeling the data distribution through a sophisticated generative procedure.

Building on this foundation, subsequent research has introduced various enhancements aimed at improving the efficiency and quality of sample generation. A key development in these variants is the introduction of adaptive noise control mechanisms, often termed noise scheduling. This control is crucial as it determines the reverse diffusion trajectory, directly influencing the fidelity and diversity of the generated samples. Among these innovations, the Score-Based Generative Model (SGM) introduced by Song & Ermon (2019); Song et al. (2020b) represents a significant advancement. SGMs utilize score-based methods, as formalized by Hyvärinen Hyvärinen & Dayan (2005), to guide the reverse diffusion. These methods leverage gradients of the data distribution to adaptively refine the generative process, producing samples that more closely resemble the original distribution. This approach has proven particularly effective in enhancing the visual and auditory quality of the generated outputs.

Another influential perspective is the treatment of diffusion as Continuous Normalizing Flows (CNFs), proposed by Lipman et al. (2022); Tong et al. (2023). This view interprets the diffusion process as a series of invertible transformations, facilitating smoother and more controlled transitions from noise back to data. This methodology is essential for maintaining the structural integrity of complex datasets and supports a more nuanced manipulation of the generative process.

Additionally, the precise control of noise levels, conceptualized through the Signal-to-Noise Ratio (S2N), has been the focus of several studies (Karras et al., 2022; Kingma et al., 2021; Kingma & Gao, 2024; Nichol & Dhariwal, 2021; Song et al., 2020a). The optimization of SNR is crucial, as it impacts the clarity and sharpness of the generated samples. By carefully tuning the SNR during the diffusion process, the model's ability to produce high-quality outputs can be significantly improved, thus avoiding common issues such as over-smoothing or excessive residual noise, which can degrade the performance of generative models. Furthermore, fast and efficient sampling has been studied in several works, notably (Song et al., 2020a; Zhang & Chen, 2022; Song et al., 2023; Zhang et al., 2023).

## 3 THEORY DEVELOPMENT

### 3.1 PROBLEM SETTING

We consider the following diffusion forward process

$$\boldsymbol{z_t} = \alpha\left(t\right)\boldsymbol{x} + \sigma\left(t\right)\boldsymbol{\epsilon},$$

where $\boldsymbol{\epsilon} \sim \mathcal{N}\left(\mathbf{0}, \mathbf{I}\right)$, $\boldsymbol{x}$ is generated from a data distribution, and $\alpha, \sigma : [0, T] \rightarrow \mathbb{R}^+$ are two functions representing signal and noise of the forward process with $\alpha\left(0\right) = 1$ and $\lim_{t \rightarrow T} \frac{\alpha(t)}{\sigma(t)} = 0$.

We define $\lambda\left(t\right) = \log \frac{\alpha(t)^2}{\sigma(t)^2}$ specifying the log of the signal-to-noise ratio with $\lim_{t \rightarrow T} \lambda\left(t\right) = -\infty$ or very low. The above signal-to-noise (S2N) forward process can be rewritten

$$z_t = \alpha\left(t\right)\boldsymbol{x} + \alpha\left(t\right)\exp\left\{-\lambda\left(t\right)/2\right\}\boldsymbol{\epsilon}, \tag{1}$$

where $\lambda\left(t\right)$ is a monotonic decreasing function from $\lambda_{max} = \lambda\left(0\right)$ and $\lambda_{min} = \lambda\left(T\right)$.

Additionally, S2N diffusion models constitute a highly diverse family of diffusion models that have achieved state-of-the-art performance in practice, as summarized in Table 1.

Table 1: Noise scheduling in various diffusion model variants.

| | $\alpha(t)$ | $\sigma(t)$ | $\lambda(t)$ | Parameters |
|---|---|---|---|---|
| VP (Song et al., 2020b) | $\frac{1}{\sqrt{e^{\frac{1}{2}\beta_d t^2 + \beta_{min} t}}}$ | $\sqrt{1 - \frac{1}{e^{\frac{1}{2}\beta_d t^2 + \beta_{min} t}}}$ | $\log \frac{1}{e^{\frac{1}{2}\beta_d t^2 + \beta_{min} t} - 1}$ | $\beta_{min} = 0.1$ $\beta_d = 19.9$ |
| VE (Song et al., 2020b) | $1$ | $\sigma_{min}(\frac{\sigma_{max}}{\sigma_{min}})^t$ | $(2t - 2)\log\sigma_{min} - 2t\log\sigma_{max}$ | $\sigma_{min} = 0.01$ $\sigma_{max} = 50$ |
| iDDPM (Nichol & Dhariwal, 2021) | $\frac{\cos(\frac{t+s}{1+s} \cdot \frac{\pi}{2})}{\cos(\frac{s}{1+s} \cdot \frac{\pi}{2})}$ | $\sqrt{1 - \frac{\cos^2(\frac{t+s}{1+s} \cdot \frac{\pi}{2})}{\cos^2(\frac{s}{1+s} \cdot \frac{\pi}{2})}}$ | $\log \frac{\cos^2(\frac{t+s}{1+s} \cdot \frac{\pi}{2})}{\cos^2(\frac{s}{1+s} \cdot \frac{\pi}{2}) - \cos^2(\frac{t+s}{1+s} \cdot \frac{\pi}{2})}$ | $s = 0.008$ |
| FM-OT (Lipman et al., 2022) | $1 - t$ | $t$ | $2\log\frac{1-t}{t}$ | |

### 3.2 THE CONNECTIVE VIEWPOINTS

**SDE Viewpoint.** From the definition of the forward process, we know that $q\left(\boldsymbol{z_t} \mid \boldsymbol{z_0}\right) = \mathcal{N}\left(\alpha\left(t\right)\boldsymbol{z_0}, \sigma^2\left(t\right)\mathbf{I}\right)$. To realize the general transition distribution $q\left(\boldsymbol{z_t} \mid \boldsymbol{z_s}\right)$ where $0 \leq s < t \leq T$, we aim to find the SDE of the above forward process. Let us consider the general form of SDE

$$d\boldsymbol{z_t} = f\left(t\right)\boldsymbol{z_t}dt + g\left(t\right)d\boldsymbol{w_t}, \tag{2}$$

where $\{\boldsymbol{w_t} : t \in [0; T]\}$ is the Brownian motion, and $f\left(t\right), g\left(t\right) \in \mathbb{R}$.

Denote $\Psi(\tau, t)$ as the transition function satisfying (i) $\frac{d\Psi(\tau,t)}{dt} = -\Psi(\tau, t) f(t) \mathbf{I}$, (ii) $\frac{d\Psi(\tau,t)}{d\tau} = \Psi(\tau, t) f(\tau) \mathbf{I}$, and (iii) $\Psi(\tau, \tau) = \mathbf{I}$. It is obvious that $\Psi(\tau, t) = \exp\left\{-\int_\tau^t f(s)\, ds\right\} \mathbf{I}$ satisfies (i), (ii), and (iii). The distribution $q(\boldsymbol{z}_t \mid \boldsymbol{z}_s) = \mathcal{N}\left(m_{t|s}, \Sigma_{t|s}\right)$ is a Gaussian distribution with $m_{t|s} = \Psi(t, s) z_s$ and $\Sigma_{t|s} = \int_s^t \Psi(t, \tau)^2 g^2(\tau)\, d\tau$. Theorem 1 whose proof can be found in Appendix .1.1 characterizes the SDE of the forward process of S2N diffusion models.

**Theorem 1** *With* $f(t) = \frac{d\log \alpha(t)}{dt} = \frac{\alpha'(t)}{\alpha(t)}$ *and* $g(t) = \sqrt{-\exp\{-\lambda(t)\}\lambda'(t)}\alpha(t)$, *the SDE flow in (2) is equivalent to the S2N forward process in (1). Moreover, we have the transition function* $\Psi(\tau, t) = \frac{\alpha(\tau)}{\alpha(t)}\mathbf{I}$ *and the transition distribution* $q(z_t \mid z_s) = \mathcal{N}\left(m_{t|s}, \Sigma_{t|s}\right)$ *with* $m_{t|s} = \frac{\alpha(t)}{\alpha(s)} z_0 = \alpha_{t|s} z_s$ *and* $\Sigma_{t|s} = \alpha^2(t)\left[\exp\{-\lambda(t)\} - \exp\{-\lambda(s)\}\right] = \alpha^2(t)\left[\frac{1}{SNR(t)} - \frac{1}{SNR(s)}\right]$ *where we define* $SNR(t) = \frac{\alpha(t)^2}{\sigma(t)^2}$. *Moreover, the SDE of the forward process of S2N diffusion models has the following form*

$$d\boldsymbol{z}_t = \frac{\alpha'(t)}{\alpha(t)}\boldsymbol{z}_t dt + \sqrt{-\exp\{-\lambda(t)\}\lambda'(t)}\alpha(t)\, d\boldsymbol{w}_t. \tag{3}$$

**Connection to continuous variational diffusion model.** Currently, our roadmap is to start from the forward SDE of S2N and derive the transition probability $q(\boldsymbol{z}_t \mid \boldsymbol{z}_s) = \mathcal{N}\left(m_{t|s}, \Sigma_{t|s}\right)$, where $m_{t|s}$ and $\Sigma_{t|s}$ are defined in Theorem 1. Interestingly, this same transition probability $q(\boldsymbol{z}_t \mid \boldsymbol{z}_s) = \mathcal{N}\left(m_{t|s}, \Sigma_{t|s}\right)$ was also used in the continuous variational diffusion model (Kingma et al., 2021) to define a continuous forward distribution for a variational approach, highlighting the connection and consistency between the forward distribution in (Kingma et al., 2021) and the forward SDE of S2N. In the next section, we aim to explore the connection between the backward SDE of S2N and the backward distribution developed in the continuous variational diffusion model (Kingma et al., 2021).

It is natural to ask a question: *if we have the transition probability* $q(\boldsymbol{z}_t \mid \boldsymbol{z}_s) = \mathcal{N}\left(m_{t|s}, \Sigma_{t|s}\right)$ *with* $m_{t|s}$ *and* $\Sigma_{t|s}$ *defined above, can we get back the SDE forward in Eq. (2)?* To this end, we consider $q(\boldsymbol{z}_{t+\Delta t} \mid \boldsymbol{z}_t)$ and derive as follows

$$\boldsymbol{z}_{t+\Delta t} = \frac{\alpha(t+\Delta t)}{\alpha(t)}\boldsymbol{z}_t + \alpha(t)\sqrt{\frac{1}{SNR(t+\Delta t)} - \frac{1}{SNR(t)}}\boldsymbol{\epsilon}$$

$$= \frac{\alpha(t+\Delta t)}{\alpha(t)}\boldsymbol{z}_t + \frac{\alpha(t)}{\sqrt{\Delta t}}\sqrt{\exp\{-\lambda(t+\Delta t)\} - \exp\{-\lambda(t)\}}(\boldsymbol{w}_{t+\Delta t} - \boldsymbol{w}_t),$$

thanks to $\boldsymbol{w}_{t+\Delta t} - \boldsymbol{w}_t = \sqrt{\Delta t}\boldsymbol{\epsilon} \sim \mathcal{N}(\mathbf{0}, \Delta t \boldsymbol{I})$.

This follows that

$$\frac{\boldsymbol{z}_{t+\Delta t} - \boldsymbol{z}_t}{\Delta t} = \frac{\alpha(t+\Delta t) - \alpha(t)}{\Delta t}\frac{\boldsymbol{z}_t}{\alpha(t)} + \alpha(t)\sqrt{\frac{\exp\{-\lambda(t+\Delta t)\} - \exp\{-\lambda(t)\}}{\Delta_t}}\frac{\boldsymbol{w}_{t+\Delta t} - \boldsymbol{w}_t}{\Delta t}.$$

By taking the limit when $\Delta t \to 0$, we obtain

$$d\boldsymbol{z}_t = \frac{\alpha'(t)}{\alpha(t)}\boldsymbol{z}_t dt + \alpha(t)\sqrt{-\exp(-\lambda(t))\lambda'(t)}d\boldsymbol{w}_t = f(t)\boldsymbol{z}_t dt + g(t)\, d\boldsymbol{w}_t,$$

which concurs with Eq. (3).

**Backward SDE.** In what follows, we examine the backward SDE of the forward SDE in Eq. (2) to indicate the connection between the backward SDE of S2N and the backward distribution developed in the continuous variational diffusion model (Kingma et al., 2021). Inspired by (Zhang & Chen, 2022), the following theorem presents the corresponding backward SDE.

**Theorem 2** *The backward SDE of the forward SDE in Eq. (2) has the following form*

$$dz_t = \left( f(t) z_t - \frac{1+\rho^2}{2} g^2(t) \nabla_{x_t} \log p(x_t) \right) dt + \rho g(t) dw_t, \tag{4}$$

*where $\rho \in \mathbb{R}$ and $\nabla_{z_t} \log p(z_t)$ is the score function. Moreover, if we use the score network $s_\theta(z_t, t)$ to estimate $\nabla_{z_t} \log p(z_t)$ and denote $s_\theta(z_t, t) = -\sigma(t)^{-1} \epsilon_\theta(z_t, t) = -\alpha(t)^{-1} \exp\left\{ \frac{\lambda(t)}{2} \right\} \epsilon_\theta(z_t, t)$, the backward SDE can be rewritten as*

$$dz_t = \left( \frac{\alpha'(t)}{\alpha(t)} z_t - \frac{1+\rho^2}{2} \exp\left\{ \frac{-\lambda(t)}{2} \right\} \lambda'(t) \alpha(t) \epsilon_\theta(z_t, t) \right) dt + \rho g(t) dw_t. \tag{5}$$

We now develop the exact solution of the backward SDE in Eq. (5), allowing us to infer or sample $z_s$ from $z_t$ with $s < t$ in Theorem 3 whose proof can be found in Appendix .1.3.

**Theorem 3** *The exact solution of the backward SDE in Eq. (5) is*

$$z_s = \frac{\alpha(s)}{\alpha(t)} z_t - \frac{1+\rho^2}{2} \alpha(s) \int_t^s \exp\left\{ \frac{-\lambda(\tau)}{2} \right\} \lambda'(\tau) \epsilon_\theta(z_\tau, \tau) d\tau$$

$$+ \rho \alpha(s) \int_t^s \sqrt{-\exp\{-\lambda(\tau)\} \lambda'(\tau)} dw. \tag{6}$$

We note that our exact solution of the backward SDE is stronger than that in (Zhang & Chen, 2022) because that work only considered the exact solution of the *backward ODE* by setting $\rho = 0$ (see Eq. (5) in (Zhang & Chen, 2022)).

Furthermore, if we consider $s = t - \Delta t$ for a small $\Delta t > 0$, we can approximate $\epsilon_\theta(z_\tau, \tau) \approx \epsilon_\theta(z_t, t)$ for $\tau \in [s, t]$, hence leading to the following approximation solution of the exact solution in Eq. (6) as shown in Corollary 1.

**Corollary 1** *If we approximate $\epsilon_\theta(z_\tau, \tau) \approx \epsilon_\theta(z_t, t)$ for $\tau \in [s, t]$, the exact solution in Eq. (6) can be approximated as*

$$z_s = \frac{\alpha(s)}{\alpha(t)} z_t + \frac{1+\rho^2}{1+\gamma} \alpha(s) \left[ \exp\left\{ \frac{-1-\gamma}{2} \lambda(t) \right\} - \exp\left\{ \frac{-1-\gamma}{2} \lambda(s) \right\} \right] \exp\left\{ \frac{\gamma \lambda(t)}{2} \right\} \epsilon_\theta(z_t, s)$$

$$+ \rho \alpha(t) \sqrt{[\exp\{-\lambda(t)\} - \exp\{-\lambda(s)\}]} \left( \frac{\alpha(s)}{\alpha(t)} \right)^{1-\delta} \left( \frac{\sigma(s)}{\sigma(t)} \right)^\delta \epsilon, \tag{7}$$

*where $\gamma \in \mathbb{R}$, $\delta \in \mathbb{R}^+$, and $\epsilon \sim \mathcal{N}(0, I)$.*

We now make connection to the backward distribution $p_\theta(z_s \mid z_t)$ developed in *continuous variational diffusion model* (Kingma et al., 2021) using the variational approach. Specifically, we have $p_\theta(z_s \mid z_t) = \mathcal{N}(z_t \mid \mu_Q(z_t; s, t), \sigma_Q^2(s, t) I)$ where we define

$$\mu_Q(z_t; s, t) = \frac{\alpha(s)}{\alpha(t)} z_t + \alpha(s) \alpha(t) (\exp\{-\lambda(t)\} - \exp\{-\lambda(s)\}) s_\theta(z_s, s)$$

$$= \frac{\alpha(t)}{\alpha(s)} z_t + \alpha(t) (\exp\{-\lambda(t)\} - \exp\{-\lambda(s)\}) \exp\left\{ \frac{\lambda(t)}{2} \right\} \epsilon_\theta(z_s, s)$$

$$\sigma_Q^2(s, t) = \sigma_{t|s}^2 \sigma_s^2 / \sigma_t^2 = \frac{\alpha(t)^2 (\exp\{-\lambda(t)\} - \exp\{-\lambda(s)\}) \sigma(s)^2}{\sigma(t)^2},$$

which is the variance of $q(z_s \mid z_t, z_0)$. This further implies that

$$z_s = \mu_Q(z_t; s, t) + \sigma_Q(s, t) \epsilon$$

$$= \frac{\alpha(s)}{\alpha(t)} z_t + \alpha(s) \alpha(t) (\exp\{-\lambda(t)\} - \exp\{-\lambda(s)\}) s_\theta(z_s, s)$$

$$+ \frac{\alpha(t) \sqrt{\exp\{-\lambda(t)\} - \exp\{-\lambda(s)\}} \sigma(s)}{\sigma(t)} \epsilon. \tag{8}$$

It is worth-noting that the inference formula in Eq. (8) is a special case of our general inference formula in Eq. (7) when $\gamma = 1$ and $\delta = 1$. This indicates that using the variational approach to obtain $p_\theta \left( z_s \mid z_t \right) = \mathcal{N} \left( z_t \mid \mu_Q \left( z_t; s, t \right), \sigma_Q^2 \left( s, t \right) \mathbf{I} \right)$, as done in *continuous variational diffusion model* (Kingma et al., 2021), falls within the spectrum of the exact solution of the corresponding backward SDE.

Moreover, the approximated inference formula in Eq. (7) is only sufficiently precise when $s = t - \Delta t$ for a small step size $\Delta t > 0$. This explains why although using $p_\theta \left( z_s \mid z_t \right)$ or the inference formula in Eq. (8) can help us to move from $z_t$ to any $z_s$ as long as $s < t$, longer step sizes $t - s$ has more errors, hence compromising the generation performance.

It is appealing to answer the question: *from the inference formula in Eq. (8), can we get back the SDE backward equation in (4) with some $\rho$?* To answer this question, from the inference formula in Eq. (8), we set $s = t - \Delta t$ to gain

$$
\frac{z_{t-\Delta t} - z_t}{-\Delta t} = \frac{\alpha \left( t - \Delta t \right) - \alpha \left( t \right)}{-\Delta t \alpha \left( t \right)} z_t
$$

$$
+ \frac{\alpha \left( t - \Delta t \right) \alpha \left( t \right) \left( \exp \left\{ -\lambda \left( t \right) \right\} - \exp \left\{ -\lambda \left( t - \Delta t \right) \right\} \right)}{-\Delta t} s_\theta \left( z_t, t \right)
$$

$$
+ \frac{\alpha \left( t \right)}{\sigma \left( t \right)} \sqrt{\frac{\exp \left\{ -\lambda \left( t \right) \right\} - \exp \left\{ -\lambda \left( t - \Delta t \right) \right\}}{\Delta t}} \sigma \left( t - \Delta t \right) \frac{w_{t-\Delta t} - w_t}{-\Delta t}.
$$

Taking limit when $\Delta t \to 0$, we gain

$$
dz_t = \left( \frac{\alpha' \left( t \right)}{\alpha \left( t \right)} z_t + \alpha^2 \left( t \right) \exp \left\{ -\lambda \left( t \right) \right\} \lambda' \left( t \right) s_\theta \left( z_t, t \right) \right) dt + \alpha \left( t \right) \sqrt{ - \exp \left\{ -\lambda \left( t \right) \right\} \lambda' \left( t \right)} dw_t
$$

$$
= \left( f \left( t \right) z_t - g \left( t \right)^2 s_\theta \left( z_t, t \right) \right) dt + g \left( t \right) dw_t,
$$

which falls in the spectrum of the SDE backward equation in (4) with $\rho = 1$. This consolidates the consistency of the SDE and the continuous variational approach viewpoints.

**Non-Markovian Continuous Variational Model and Its SDE.** Inspired by DDIM (Song et al., 2020a), we relax the Markov property in the forward process and aim to find the backward distribution $q(z_s | z_t, x)$ ($s < t$) such that its induced marginal distribution $q(z_s)$ coincides with the forward one. To achieve this, we consider

$$
q \left( z_s \mid z_t, x \right) = \mathcal{N} \left( \alpha \left( s \right) z_0 + \sqrt{\sigma^2 \left( s \right) - \beta^2 \left( s, t \right)} \frac{z_t - \alpha \left( t \right) z_0}{\sigma \left( t \right)}, \beta^2 \left( s, t \right) \mathbf{I} \right).
$$

Here we note that different from DDIM (Song et al., 2020a) which aims to characterize the *discrete* Non-Markovian backward distribution $q \left( z_{t-1} \mid z_t, x \right)$, we aim to characterize the *continuous* Non-Markovian backward distribution, allowing us to jump backward from $z_t$ to $z_s$ with $s < t$. One might argue that for DDIM (Song et al., 2020a), in $q \left( z_{t-1} \mid z_t, x \right)$, we can replace $z_{t-1}$ by $z_{t-\Delta t}$ ($\Delta t \to 0$) to reach the result for the continuous flow. However, it is not trivial since it requires to perform an asymptotic analysis as done in Kingma et al. (2021).

We need to prove the *consistency* between the forward and backward processes (i.e., the induced marginal distribution $q(z_s)$ coincides with the forward one). Indeed, we prove by induction, i.e., if $q \left( z_t \mid x \right) = \mathcal{N} \left( \alpha \left( t \right) x, \sigma(t)^2 \mathbf{I} \right)$ then $q \left( z_s \mid x \right) = \mathcal{N} \left( \alpha \left( s \right) x, \sigma^2 \left( s \right) \mathbf{I} \right)$. This is obvious because we have

$$
q \left( z_s \mid x \right) = \int q \left( z_s \mid z_t, x \right) q \left( z_t \mid x \right) dz_t,
$$

and from Bishop (2007), the mean and variance of $z_s$ can be computed as

$$
m \left( z_s \right) = \alpha \left( s \right) x + \sqrt{\sigma^2 \left( s \right) - \beta^2 \left( s, t \right)} \frac{\alpha \left( t \right) x - \alpha \left( t \right) x}{\sigma \left( t \right)} = \alpha \left( s \right) x.
$$

$$
V \left( z_s \right) = \beta^2 \left( s, t \right) \mathbf{I} + \left( \sigma^2 \left( s \right) - \beta^2 \left( s, t \right) \right) \frac{\sigma^2 \left( t \right)}{\sigma^2 \left( t \right)} \mathbf{I} = \sigma^2 \left( s \right) \mathbf{I}.
$$

Given $q\left(\boldsymbol{z}_T \mid \boldsymbol{x}\right) = \mathcal{N}\left(\alpha\left(T\right)\boldsymbol{x}, \sigma^2\left(T\right)\mathbf{I}\right)$, we reach $q\left(\boldsymbol{z}_t \mid \boldsymbol{x}\right) = \mathcal{N}\left(\alpha\left(t\right)\boldsymbol{x}, \sigma^2\left(t\right)\mathbf{I}\right), \forall t \in [0; T]$, indicating that $q\left(\boldsymbol{z}_s \mid \boldsymbol{z}_t, \boldsymbol{x}\right) = \mathcal{N}\left(\alpha\left(s\right)\boldsymbol{z}_0 + \sqrt{\sigma^2\left(s\right) - \beta^2\left(s,t\right)}\frac{\boldsymbol{z}_t - \alpha(t)\boldsymbol{z}_0}{\sigma(t)}, \beta^2\left(s,t\right)\mathbf{I}\right)$ is a proper backward flow.

Moreover, by defining

$$p_\theta\left(\boldsymbol{z}_s \mid \boldsymbol{z}_t\right) = q\left(\boldsymbol{z}_s \mid \boldsymbol{z}_t, \hat{z}_\theta\left(\boldsymbol{z}_t, t\right)\right),$$

where $\hat{z}_\theta\left(\boldsymbol{z}_t, t\right) = \frac{\sigma^2(t)s_\theta(\boldsymbol{z}_t,t) + \boldsymbol{z}_t}{\alpha(t)} = \frac{\boldsymbol{z}_t - \sigma(t)\epsilon_\theta(\boldsymbol{z}_t,t)}{\alpha(t)}$ is used to predict $\boldsymbol{x}$, we reach

$$\boldsymbol{z}_s = \alpha\left(s\right)\hat{z}_\theta\left(\boldsymbol{z}_t, t\right) + \sqrt{\sigma^2\left(s\right) - \beta^2\left(s,t\right)}\frac{\boldsymbol{z}_t - \alpha\left(t\right)\hat{z}_\theta\left(\boldsymbol{z}_t, t\right)}{\sigma\left(t\right)} + \beta\left(s,t\right)\boldsymbol{\epsilon}$$

$$= -\left(\exp\left\{\frac{\lambda\left(s\right)}{2}\right\} - \exp\left\{\frac{\lambda\left(t\right)}{2}\right\}\sqrt{1 - \frac{\beta^2\left(s,t\right)}{\sigma^2\left(s\right)}}\right)\exp\left\{\frac{-\lambda\left(t\right) - \lambda\left(s\right)}{2}\right\}\alpha\left(s\right)\epsilon_\theta\left(\boldsymbol{z}_t, t\right)$$

$$+ \frac{\alpha\left(s\right)}{\alpha\left(t\right)}\left(1 + \exp\left\{\frac{\lambda\left(t\right) - \lambda\left(s\right)}{2}\right\}\sqrt{1 - \frac{\beta^2\left(s,t\right)}{\sigma^2\left(s\right)}}\right)\boldsymbol{z}_t + \beta\left(s,t\right)\boldsymbol{\epsilon}, \tag{9}$$

where $\boldsymbol{\epsilon} \sim \mathcal{N}(\mathbf{0}, \mathbf{I})$.

It is worth-noting that Eq. (9) enables us to jump from $\boldsymbol{z}_t$ to $\boldsymbol{z}_s$ ($s < t$). Moreover, this can be considered as a continuous and generalizing version of Eq. (12) in DDIM (Song et al., 2020a). More interestingly, in the following theorem, we find out the SDE that corresponds to the continuous Non-Markovnian variational model in Eq. (9).

**Theorem 4** *Consider $\beta\left(s,t\right) = \sqrt{b\left(s\right) - b\left(t\right)}$ for $s < t$ with a decreasing function $b$. The SDE that corresponds to the continuous variational model in Eq. (9) has the following form:*

$$d\boldsymbol{z}_t = \left[\frac{\alpha'\left(t\right)}{\alpha\left(t\right)} + \frac{\lambda'\left(t\right)}{2}\exp\left\{\frac{\lambda\left(t\right)}{2}\right\}\right]\boldsymbol{z}_t - \frac{\lambda'\left(t\right)}{2}\exp\left\{-\frac{\lambda\left(t\right)}{2}\right\}\alpha\left(t\right)\epsilon_\theta\left(\boldsymbol{z}_t, t\right)dt$$

$$+ \frac{1}{2}b'\left(t\right)\exp\left\{\frac{\lambda\left(t\right)}{2}\right\}\alpha^{-1}\left(t\right)\epsilon_\theta\left(\boldsymbol{z}_t, t\right) + \sqrt{-b'\left(t\right)}d\boldsymbol{w}_t. \tag{10}$$

Here we note that the result obtained in Theorem 4 is totally novel because what was gained in Song et al. (2020a) is the connection between the DDIM iterate and the probability flow ODE (Song et al., 2020b).

### 3.3 TRANSFORMING S2N DIFFUSION MODELS TO SIGNAL-TO-NOISE SPACE AND INFORMATION THEORY VIEWPOINT

Similar to Kingma et al. (2021); Kingma & Gao (2024), we transform the S2N diffusion models to the signal-to-noise space that enables us to investigate the *information-theoretic viewpoint* of the S2N diffusion models.

We denote $\tilde{\alpha}\left(\lambda\left(t\right)\right) = \alpha\left(t\right)$ (i.e., $\tilde{\alpha} = \alpha \circ \lambda^{-1}$), $\tilde{\sigma}\left(\lambda\left(t\right)\right) = \sigma\left(t\right)$ (i.e., $\tilde{\sigma} = \sigma \circ \lambda^{-1}$), and $\tilde{\boldsymbol{z}}_{\lambda(t)} = \boldsymbol{z}_t$ where $\lambda\left(t\right) = \log\frac{\alpha(t)}{\sigma(t)} = \log\frac{\tilde{\alpha}(\lambda(t))^2}{\tilde{\sigma}(\lambda(t))^2}$. We have the following forward process in the signal-to-noise space

$$\tilde{\boldsymbol{z}}_{\lambda(t)} = \tilde{\alpha}\left(\lambda\left(t\right)\right)\boldsymbol{x} + \tilde{\sigma}\left(\lambda\left(t\right)\right)\boldsymbol{\epsilon} = \tilde{\alpha}\left(\lambda\left(t\right)\right)\boldsymbol{x} + \frac{\tilde{\alpha}\left(\lambda\left(t\right)\right)}{\exp\left\{\lambda\left(t\right)/2\right\}}\boldsymbol{\epsilon}$$

or equivalently

$$\tilde{\boldsymbol{z}}_\lambda = \tilde{\alpha}\left(\lambda\right)\boldsymbol{x} + \frac{\tilde{\alpha}\left(\lambda\right)}{\exp\left\{\lambda/2\right\}}\boldsymbol{\epsilon}, \tag{11}$$

where $\lambda \in [\lambda_{min}, \lambda_{max}]$ with $\lambda_{min} = \lambda\left(T\right)$ and $\lambda_{max} = \lambda\left(0\right)$.

In the following theorem, we answer the question which pair of $\left(\alpha\left(t\right), \sigma\left(t\right)\right)$ induces the same forward process in the signal-to-noise space.

**Theorem 5** *Given $\left(\alpha_1\left(t\right), \sigma_1\left(t\right)\right)$, $\sigma_1 \circ \lambda_1^{-1} = \sigma_2 \circ \lambda_2^{-1}$, and $\left(\alpha_2\left(t\right), \sigma_2\left(t\right)\right)$, if $\lambda_1\left(0\right) = \lambda_2\left(0\right)$, $\lambda_1\left(T\right) = \lambda_2\left(T\right)$, and $\alpha_1 \circ \lambda_1^{-1} = \alpha_2 \circ \lambda_2^{-1}$, the forward processes corresponding to $\left(\alpha_1\left(t\right), \sigma_1\left(t\right)\right)$ and $\left(\alpha_2\left(t\right), \sigma_2\left(t\right)\right)$ induce the same forward process in the signal-to-noise space.*

Theorem 5 indicates that some S2N diffusion models induce the same diffusion model on the signal-to-noise space. Moreover, by defining the corresponding relation, we can group the S2N DMs that induce the same diffusion model on the signal-to-noise space in the equivalent classes.

**Information-Theoretic viewpoint of S2N DM in the signal-to-noise space.** Information theoretic viewpoint was studied in Kong et al. (2023) for a very simple diffusion process: $\tilde{z}_\lambda = \sqrt{\lambda}x + \epsilon$. It is appealing to generalize this information theoretic result for a general and more practical diffusion process. In what follows, we develop the information-theoretic results for the general S2N diffusion model in the signal-to-noise space in Eq. (11).

Given $x$, we define the Minimum Mean Square Error (MMSE) for recovering $x$ in the noisy channel

$$\mathrm{mmse}\,(\lambda) := \min_{\hat{x}(\tilde{z}_\lambda, \lambda)} \mathbb{E}_{p(\tilde{z}_\lambda, x)}\left[\|x - \hat{x}\,(\tilde{z}_\lambda, \lambda)\|_2^2\right],$$

where $\hat{x}\,(\tilde{z}_\lambda, \lambda)$ is referred to as a denoising function. The optimal denoising function $\hat{x}^*$ corresponds to the conditional expectation, which can be seen using variational calculus or from the fact that the squared error is a Bregman divergence

$$\hat{x}^*\,(\tilde{z}_\lambda, \lambda) = \mathrm{argmin}_{\hat{x}(\tilde{z}_\lambda, \lambda)}\mathbb{E}_{p(\tilde{z}_\lambda, x)}\left[\|x - \hat{x}\,(\tilde{z}_\lambda, \lambda)\|_2^2\right] = \mathbb{E}_{x\sim p(x|z_\lambda)}\,[x].$$

Moreover, the point-wise MMSE is defined as follows:

$$\mathrm{mmse}\,(x, \lambda) := \mathbb{E}_{p(\tilde{z}_\lambda, x)}\left[\|x - \hat{x}^*\,(\tilde{z}_\lambda, \lambda)\|_2^2\right].$$

The mutual information $\mathbb{I}\,(x, \tilde{z}_\lambda)$ can be characterized in the following theorem.

**Theorem 6** *For a general S2N DM in the general signal-to-noise space, we have*

*(i)* $\frac{d}{d\lambda}D_{KL}\,(p\,(\tilde{z}_\lambda \mid x)\,\|p\,(\tilde{z}_\lambda)) = -\frac{D\tilde{\sigma}'(\lambda)}{2\tilde{\sigma}(\lambda)} + \frac{[\tilde{\alpha}'(\lambda)\tilde{\sigma}(\lambda) - \tilde{\alpha}(\lambda)\tilde{\sigma}'(\lambda)]\tilde{\alpha}(\lambda)}{\tilde{\sigma}^3(\lambda)}mmse\,(x, \lambda)$ *where $D$ is the dimension of $\tilde{z}_\lambda$, $D_{KL}$ is the Kullback-Leibler divergence, and $\tilde{\sigma}\,(\lambda) = \tilde{\alpha}\,(\lambda)\exp\{-\lambda/2\}$.*

*(ii)* $\frac{d}{d\lambda}\mathbb{I}\,(x, \tilde{z}_\lambda) = -\frac{D\tilde{\sigma}'(\lambda)}{2\tilde{\sigma}(\lambda)} + \frac{[\tilde{\alpha}'(\lambda)\tilde{\sigma}(\lambda) - \tilde{\alpha}(\lambda)\tilde{\sigma}'(\lambda)]\tilde{\alpha}(\lambda)}{\tilde{\sigma}^3(\lambda)}mmse\,(\lambda).$

It is worth noting that our results obtained in Theorem 6 can lead to those in Kong et al. (2023) when choosing $\tilde{\alpha}\,(\lambda) = \sqrt{\lambda}$ and $\tilde{\sigma}\,(\lambda) = 1$.

## 4 EXPERIMENTS

Inspired by the theoretical results in Section 3, we conduct experiments to test the effectiveness of hyperparameters participating in the backward process built based on our Corollary 1. Our experiment settings are organized based on work and checkpoints in EDM (Karras et al., 2022).

### 4.1 DETERMINISTIC SAMPLING

Corollary 1 becomes deterministic when $\rho = 0$, then the sampling process is defined as follow:

$$z_s = \frac{\alpha(s)}{\alpha(t)}z_t - \frac{1}{1+\gamma}\alpha(s)\left[\exp\left\{\frac{-1-\gamma}{2}\lambda(t)\right\} - \exp\left\{\frac{-1-\gamma}{2}\lambda(s)\right\}\right]\exp\left\{\frac{\gamma\lambda(t)}{2}\right\}\epsilon_\theta\,(z_t, s) \tag{12}$$

The traditional Euler solver method corresponds to our specific case when $\gamma = 0$. As shown in Figure 4, with the same number of NFEs, more negative $\gamma$ makes the outcome images blurrier, while images become sharper as $\gamma$ increases. However, too large a $\gamma$ value exceeds the common range of pixel values and distorts the images.

Figure 1 represents our grid search results to find the optimal value of $\gamma$ for each CIFAR-10 $(32\times32)$ model pretrained by Song et al. (2020b) and Karras et al. (2022). We observe that in all settings, the optimal $\gamma$ value, which corresponds to the best FID, is a small positive number, especially around 0.026 for the settings used by Karras et al. (2022). Not only for CIFAR-10 $(32 \times 32)$, but $\gamma = 0.026$

Table 2: Results in FID ($\downarrow$) for Unconditional FFHQ ($64 \times 64$), Unconditional AFHQv2 ($64 \times 64$) and Conditional ImageNet ($64 \times 64$) settings by deterministic sampling with NFE = 79 using Karras et al. (2022)'s checkpoints.

|  | Uncond. FFHQ | | Uncond. AFHQv2 | | Cond. ImageNet |
| --- | --- | --- | --- | --- | --- |
|  | VP | VE | VP | VE | |
| Euler solver | **3.25** | 3.43 | 2.38 | 2.58 | 2.75 |
| Ours ($\gamma = 0.026$) | 3.27 | **3.39** | **2.09** | **2.27** | **2.71** |

also outperforms $\gamma = 0$ in nearly all cases for the Unconditional FFHQ ($64 \times 64$), Unconditional AFHQv2 ($64 \times 64$), and Conditional ImageNet ($64 \times 64$), as shown in Table 2. With the FFHQ dataset, we achieve only an approximate result in VP ($3.27 > 3.25$) and a slight improvement in VE ($3.39 < 3.43$). Results in ImageNet also display slight improvement ($2.71 < 2.75$). On the other hand, the evaluation for AFHQv2 shows a significant decrease in FID: 0.29 in VP and 0.31 in VE.

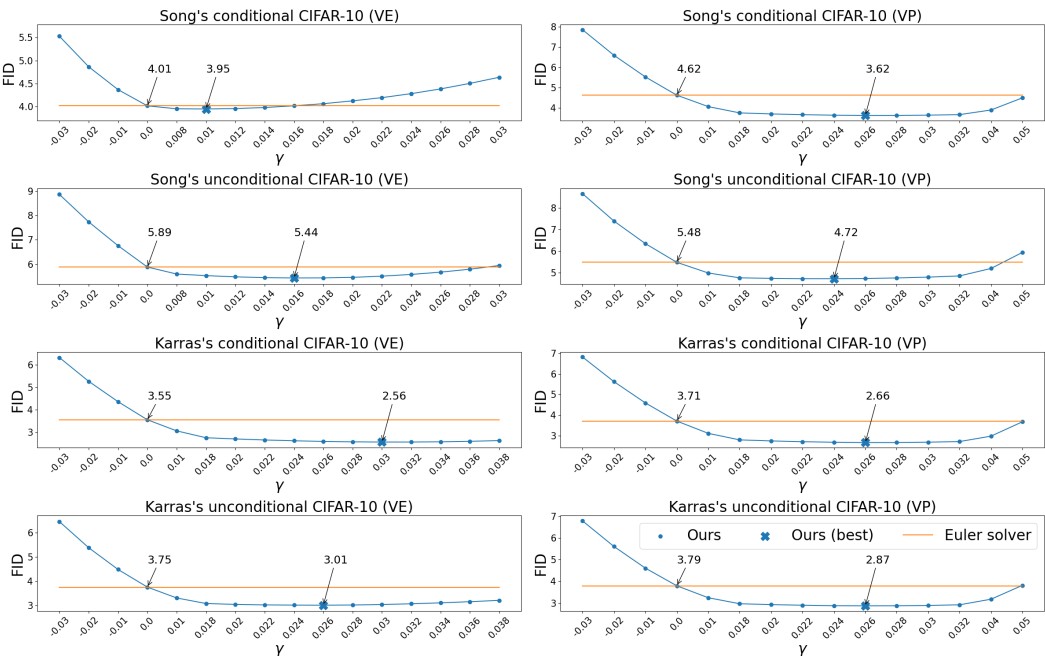

Figure 1: The FID ($\downarrow$) of deterministic sampling at several CIFAR-10 ($32 \times 32$) checkpoints when varying $\gamma$ with NFE = 35.

## 4.2 STOCHASTIC SAMPLING

We observe synthetic image quality in stochastic sampling ($\rho = 1$) for numerous values of $\gamma$ and $\delta$. As mentioned before, setting $\gamma = 1$ and $\delta = 1$ allows Corollary 1 to correspond to the traditional inference formula 8. Similarly to deterministic sampling 4.1, different choices of $\gamma$ and $\delta$ can improve results when evaluating model performance. Figure 3 presents the FID for an unconditional CIFAR-10 ($32 \times 32$) model from a grid search over $\gamma$ and $\delta$. Among the pool of candidates, the choice of ($\gamma = 1.25, \delta = 0.95$) yields the best value for this metric and improves model performance beyond ($\gamma = 1, \delta = 1$), not only in this CIFAR-10 setting but also in an ImageNet ($64 \times 64$) setting, as shown in Figure 2.

## 5 CONCLUSION

Diffusion models (DMs) have emerged as essential elements within generative models, demonstrating proficiency across diverse domains such as image synthesis, audio generation, and intricate data

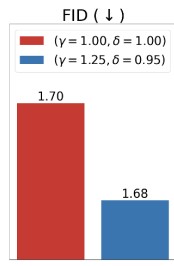
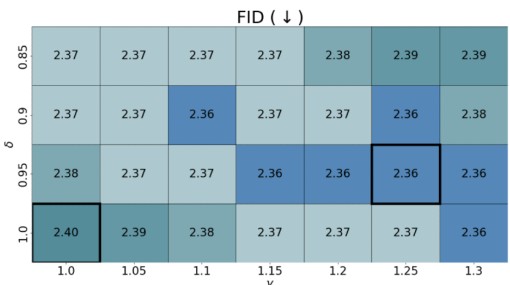

Figure 2: FID ($\downarrow$) for Class-Conditional ImageNet ($64 \times 64$) dataset by stochastic sampling with NFE = 511.

Figure 3: Grid search results in FID ($\downarrow$) for Unconditional CIFAR-10 ($32 \times 32$) (VP) by stochastic sampling when varying $\gamma$, $\delta$ with NFE = 511.

interpolation. Signal-to-Noise diffusion models encompass a versatile family that includes most cutting-edge diffusion models. While various efforts have been made to analyze Signal-to-Noise (S2N) diffusion models from different angles, there is still a need for a comprehensive investigation that connects disparate perspectives and explores novel viewpoints. In this work, we present an extensive examination of noise schedulers, probing their significance through the prism of the signal-to-noise ratio (SNR) and its links to information theory. Expanding upon this framework, we have devised a generalized backward equation aimed at enhancing the efficacy of the inference process. Our experimental results show that by choosing the correct hyperparameters, our generalized equation improves model performance compared to traditional ones.

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
