# OpenReview forum: "Unified Perspectives on Signal-to-Noise Diffusion Models"
_ICLR.cc/2025/Conference — ICLR 2025 Conference Withdrawn Submission_

### Official Review · Reviewer_zsuQ · 2024-10-28

**Soundness:** 3
**Presentation:** 3
**Contribution:** 3
**Rating:** 5
**Confidence:** 5

**Summary:**

The paper investigates the role of noise schedulers in diffusion models and examines their relationship with Signal-to-Noise (S2N) diffusion models and concepts from information theory. Additionally, it introduces a generalized backward equation aimed at enhancing the efficiency of the inference process.

**Strengths:**

1. The paper show how the forward SDE of S2N diffuson model is related to Kingma et al. (2021) diffusion model.
2. develop a continuous variational diffusion model capable of exactly inducing
the forward distributions.
3.  develop an information-theoretic perspective for a general S2N diffusion model
4. achieve higher performance than the inference baselines

**Weaknesses:**

1. The reported improvements in performance, such as slight reductions in FID scores, are minimal in many cases (e.g., 3.39 vs. 3.43 on FFHQ). These marginal gains raise concerns about whether the proposed methodologies justify the increased complexity introduced by the framework.

2. The experimental validation relies on relatively low-resolution datasets, including CIFAR-10 (32×32), FFHQ (64×64), AFHQv2 (64×64), and ImageNet (64×64). The limited resolution restricts the generalizability of the findings, as the proposed methods are not tested on higher-resolution or more complex datasets.

3. The paper is heavily builds on prior work by Kingma et al. (2021) and Song et al. (2020a). The reported advancements offer only minor improvements over these existing frameworks, leading to questions about the novelty and impact of the contribution.

**Questions:**

--

---

### Official Review · Reviewer_V6ok · 2024-11-01

**Soundness:** 3
**Presentation:** 3
**Contribution:** 2
**Rating:** 5
**Confidence:** 3

**Summary:**

The authors show how to derive a generalized representation from a forward SDE based on S2N diffusion and its backward process that can include continuous VDM, DEIS as a special case. Furthermore, they extend the continuous VDM to derive exact solutions for the forward-backward process and the backward SDE under a continuous Non-Markovian distribution. Third, the authors show that an information theoretic representation of S2N DMs naturally yields a representation that includes the work of Kong et al (2023) as a special case. Finally, the authors show through a simple experiment that better hyperparameter search is possible based on this generalized expression.

**Strengths:**

- Overall, the paper is well written, easy to follow, and the arguments are logical.
- The main contribution is to provide an overall understanding of the DM approaches by providing a unifying view of the various DMs from different perspectives.
    - Development of a forward Stochastic Differential Equation (SDE) for S2N diffusion models and demonstration of its consistency with VDM.
    - Extending VDM for a continuous Non-Markovian process
    - Transformation of S2N diffusion models into the signal-to-noise space, enabling an information-theoretic perspective.
- Another contribution is the didactic value of the detailed explanations throughout the first 8 pages. This paper is a great way to summarize a lot of knowledge about diffusion that is often scattered among many papers (and sometimes disconnected from each other, making it difficult for readers to get it all at once.)

**Weaknesses:**

- The main weakness of the paper is the lack of the authors' insights on the benefits of using the theoretical contributions and the lack of experimental validation of them.
- At this point, analysis of experimental results is lacking in some respects. For now, it looks like Section 4 is just paraphrasing facts that can be seen in the figures.
    - Could you provide a more detailed explanation of Figures 1, 2, and 3? Is the existence of a better hyperparameter an unconditional tendency over datasets/models? If not, what kind of conditions apply?
    - In theory, many previous studies have shown that this can be a special case, and as far as I know, those studies (e.g. VDM++) have reported performance on high resolution data. Are there any experimental results and analysis of hyperparameter search on high-resolution data?

**Questions:**

- What kind of performance gains would be possible by getting a unified viewpoint and generalized expression (other than acquiring better hyperparameter)?
- Is it possible to do a hyperparameter search other than a grid search (e.g., adaptive scheduling)?
- Minor comment
    - The parenthetical abbreviation for S2N should appear on L15.
    - The parenthetical abbreviation for S2N should not appear on L114.
    - The authors’ effort, to explain the theoretical developments as kindly as possible, is highly appreciated. But I think there's too much content in Section 3 and too little in Section 4, so I would suggest moving some of the content in Section 3 to the Appendix. In particular, Theorem 2 and Theorem 5 could be written as propositions in the appendix, rather than presenting them in the text as theorems.

---

### Official Review · Reviewer_sQTi · 2024-11-03

**Soundness:** 3
**Presentation:** 1
**Contribution:** 2
**Rating:** 3
**Confidence:** 3

**Summary:**

The present paper investigates signal-to-noise diffusion models from a theoretical perspective. It chiefly connects three prior works: from Kingma et al. (2021), Zhang & Chen (2022), and Song et al. (2020), as well as several other seminal works on diffusion models. The work aims to offer theoretical insights into these models from a connective viewpoint, and results in two derived inference formulae with novel hyperparameters. The authors run experiments using several pretrained diffusion models to compare the choice of these hyperparameters and find improvements in FID scores from nonstandard choices.

**Strengths:**

The work is original and significant in its explicit linking between several prior theoretical works and viewpoints on diffusion models. The mathematical derivations seem correct as far as I can tell, and the English writing quality is good. The results in Figure 1 are interesting and show that most existing models may not make optimal sampler choices for achieving low FID scores, at least when specifically considering Euler-style first-order solvers.

**Weaknesses:**

* In its present form, the paper is not well-organized and reads disjointed. Without having intimate knowledge of each cited work, it is at many points unclear what parts are repeated from prior work and what is actually novel, since reproduced and (apparently) newly derived formulae are intermixed in writing without clear distinctions. For example, Theorem 2 / Eq. (4) is reproduced from Zhang & Chen (2022, Eq. 16), but is not clearly marked as such, instead it is only noted as being "inspired by (Zhang & Chen, 2022)", and the following formula Eq. (5) seems to be simple substitution of the authors' SNR-based expression, as far as I can tell. **This could be improved by stating clearly which theoretical results are reproduced and which are novel, and what the motivation and purpose of extending, generalizing, or modifying each known result is.**

* Furthermore, it is unclear what the goal of each section is, as many subsections of the main section 3 do not seem to be connected and do not form a cohesive story. Specifically, the authors derive (or reproduce) six theorems and one corollary in section 3, but in section 4, they only make use of the corollary (which is a corollary of theorem 3) to perform any experiments. This makes it unclear what the purpose of theorems 4, 5, and 6 in this work is, especially since they also do not seem to be connected to each other. **This could be improved by rewriting sections towards a more coherent story linking the results, and by adding experiments related to theorems 4-6, if possible.** For example, for theorem 5, the authors could actually list a grouping of S2N models that "induce the same diffusion model on the signal-to-noise space", as stated as a possibility in lines 378-380. For theorem 6, the authors could attempt an empirical comparative analysis of the mutual information of various S2N models, e.g. the pretrained diffusion models used for the other experiments.

* The SNR-based viewpoint, which according to the title should be a key feature of the paper, does not seem really necessary for theorems 1 to 3 and corollary 1 (which, as noted, are however the only theoretical results that are used in the experimental section), unless I am missing something. This viewpoint - as used in this work - also seems to generally increase the complexity of the involved expressions rather than reduce it, whereas the reduction of complexity was one key advantage of originally introducing this viewpoint in Kingma et al. (2022). **Please clarify the use of this viewpoint for theorems 1-3 and corollary 1.**

Reviewer comment: The SNR-based viewpoint, which according to the title should be a key feature of the paper, does not seem really necessary for theorems 1 to 3 and corollary 1 (which, as noted, are however the only theoretical results that are used in the experimental section). This viewpoint - as used in this work - also seems to generally increase the complexity of the involved expressions rather than reduce it, whereas the reduction of complexity was one key advantage of originally introducing this viewpoint in Kingma et al. (2022). Could the authors clarify how the SNR-based perspective contributes to each theorem and the corollary?

* Some expressions and derivations listed in the main text seem overly extensive, showing trivial steps that may better be left in the Appendix, for example in lines 253-269. Doing so could free up space for writing a more cohesive storyline and for easier understanding of the key contributions.

* The experimental section seems highly limited in scope and generalizability:
  * The authors only evaluate a small selection of models and use FID as the single metric to compare models. For models based on the datasets FFHQ and the conditional model of ImageNet, only marginal differences in FID are found (Table 2). Only for AFHQ, the proposed choice $\gamma = 0.026$ seems to make a real difference for FID. For CIFAR-10 32x32, the gains from $\gamma \neq 0$ are more clear (Fig. 1), but this dataset is very limited by its resolution and size, making it generally unsuitable for drawing deep conclusions.
  * The best choice of the hyperparameters $\gamma, \delta$ (and perhaps $\rho$) may well be very different for other datasets or even different network architectures, but the experimental section lacks any discussion of this.
  * The results in Figure 2 and 3, setting $\rho = 1$ (as in standard SDE sampling) and varying $\delta, \gamma$ are not significant at all, with an FID score of 1.68 vs. 1.70 for conditional ImageNet and between 2.36 and 2.40 for unconditional CIFAR-10.
  * This seems to suggest that, beyond results on the limited CIFAR-10 set (Figure 1), there are no relevant gains from the proposed sampler, and it requires expensive grid-searches for up to three hyperparameters, severely limiting its practical usefulness.
  * **The trustworthiness of the presented empirical results could be improved by evaluating on other datasets or tasks, other pretrained models (not only those trained with the EDM formalism) and/or comparing against other samplers proposed in the literature, at similar NFE.**

**Questions:**

* $\gamma, \delta$ and $\rho$ seem to be introduced without a motivation. Can the authors provide a deeper intuition for the meaning of these hyperparameters?

* In the experimental section, the authors evaluate the parameters $\gamma$ and $\delta$ from Corollary 1 through line and grid searches, but evaluate only for $\rho$ of either 0 or 1 - this has already been done in many other works (when comparing Probability Flow ODE samplers $\rho = 0$ and SDE samplers $\rho = 1$). Is there any reason why the authors did not also investigate other choices of $\rho$ here?

* The first paragraph of Section 2 seems a highly redundant reproduction of the first paragraph of Section 1. I would suggest either adding additional relevant citations here to make this paragraph more relevant in a "Related Work" section, or to shorten it to reduce the redundancy.

* $\beta(s,t)$ is first used on page 6, lines 304-306, but seems to only be defined on page 7, in Theorem 4. Please introduce $\beta$ where it is first used.

* In Figure 2, why did the authors choose to show the two numbers on a bar chart without a proper y-axis that clearly does not start at 0? Comparing the two values 1.68 and 1.70 does not call for a bar chart, and this figure falsely visually indicates a much larger difference than the actual numerical difference (0.02). I would suggest removing this figure altogether as it is not helpful, and its results can be stated in the main text, or in a small table.

* In section 4.2 (Figures 2 and 3), why did the authors choose to evaluate for an NFE of 511, rather than consistently choosing NFE=35 as done in section 4.1 (Figure 1)?

Typos/typesetting:

* "Non-Markovnian" -> "Non-Markovian" (this typo is repeated multiple times from line 300 onwards, please correct each occurrence)
* I would suggest using "\text{SNR}" instead of "SNR" in math mode, as just "SNR" (e.g. in lines 194-197 or 173-174) reads as three disjoint symbols rather than the symbol for a single operator.
* Please consistently use equation numbers throughout the paper and do not omit them (e.g. lines 194-210); this would make referring to each part for discussions easier.

---

### Official Review · Reviewer_EibK · 2024-11-04

**Soundness:** 2
**Presentation:** 1
**Contribution:** 1
**Rating:** 3
**Confidence:** 4

**Summary:**

This study aims to unify and organize existing diffusion model techniques from a theoretical perspective, providing a cohesive framework and attempting to derive new insights.

**Strengths:**

- An attempt to integrate/organize existing diffusion models from a theoretical perspective. It is actually a significant attempt. However, the originality and clarity of the theory are considered to be limited (see below).

**Weaknesses:**

- Despite the claim of presenting a comprehensive framework, the scope of the work appears quite limited. While the paper aims to present a unified perspective, the introduction of numerous variables and complicated notation makes the theoretical framework difficult to follow. Unfortunately, it does not seem to succeed in building a clear and cohesive theory. Additionally, there appear to be several similar studies that are not adequately referenced.

- At first glance, the work appears mathematically intensive, but the complexity seems to stem mainly from cumbersome notation and calculations rather than any mathematically nontrivial arguments. Decorative math without substantial purpose is unnecessary.
  - *Edited after Chair's suggestions*
    - For instance, SNR is defined as $\lambda(t)$, yet the paper also introduces another notation, $SNR(t)$. This redundancy adds unnecessary complexity and increases the cognitive load on the reader.
    - The origin or derivation of the transition function $\Phi(\tau, t)$ is unclear and needs further explanation.
    - Theorem 1 appears too trivial, serving merely as an introduction of notation. It seems to be a simple variable transformation and does not warrant being presented as a theorem. Instead, it could be streamlined into the main text without the excessive formality. The same critique applies to Theorem 2, which merely rephrases known results using the authors' notation, lacking any substantive contribution.
    - Another example is the discussion that describes the Euler-Maruyama method with time discretization. It would suffice to state that an SDE of the form $dz_t = f_t dt + g_t dB_t$ is discretized as $z_{t+1} - z_t = f_t \Delta t + g_t (w_{t+1} - w_t)$.
    - The authors claim that Theorem 3 provides a stronger result compared to existing work; however, the improvement is marginal and lacks nontrivial insights. The parameter $\rho$ is introduced, yielding a known result when $\rho = 0$ or $\rho = 1$. Yet, this formulation appears straightforward if one is familiar with the derivation process. In fact, I once have derived a similar formula several years ago. This generalization is not particularly challenging, and the significance of this generalization remains unclear in the paper, as it only elaborates on the specific cases of $\rho = 0$ and $\rho = 1$.
    - Despite the formality, the definition of *signal-to-noise space* is not given, which the authors claim the contribution of the paper.
    - Overall, the paper contains an abundance of mathematical ornamentation, where existing results are merely rephrased in the authors' notation. It is difficult to identify any substantial contributions, and the writing style would benefit from greater clarity and conciseness.

- The number of cited references is notably low compared to similar papers, and several important works on diffusion models are missing. In addition, more thorough citation of fundamental literature on SDEs, stochastic process or nonequilibrium thermodynamics would be appropriate. These references are arguably more important than citations like Bishop or Hyvarinen for the context of this work.
  - *Edit after Chair's suggestions*
    - For example, relevant literature can be found in the survey compiled at [https://github.com/chq1155/A-Survey-on-Generative-Diffusion-Model]. Some of the theoretical literature will be closely related to this paper.
    - See also https://arxiv.org/abs/2209.00796.
    - I think that Ho's work already provides a comprehensive general framework.


- The example images presented in the appendix might have been compelling in 2020, but in 2025, they no longer offer anything particularly interesting or novel.

**Questions:**

- The study mentions using grid search for parameter tuning, but it is unclear whether the dataset was properly divided into separate sets for parameter search and validation, or if the same data was used for both. This distinction needs to be clarified.
- It is also unclear why the baseline method is chosen to be Euler's method, as it seems clearly inappropriate for a meaningful comparison, as there already exist many higher-order samplers (Figure 1). Also, while the paper claims a slight performance improvement over Euler's method, the improvement appears too marginal to determine statistical significance. It would be helpful to provide confidence intervals or similar measures to support this claim (Table 2).

---

### Note · Authors · 2024-12-02

I have read and agree with the venue's withdrawal policy on behalf of myself and my co-authors.